# Observation of novel charge ordering and spin reorientation in perovskite oxide PbFeO$_3$

Xubin Ye [1,2,14], Jianfa Zhao [1,2,14], Hena Das [3,4,14], Denis Sheptyakov [5], Junye Yang [5], Yuki Sakai [6,3], Hajime Hojo [7], Zhehong Liu [1,2], Long Zhou [1,2], Lipeng Cao [1], Takumi Nishikubo [3], Shogo Wakazaki [3], Cheng Dong [1,2], Xiao Wang [8], Zhiwei Hu [8], Hong-Ji Lin [9], Chien-Te Chen [9], Christoph Sahle [10], Anna Efiminko [10], Huibo Cao [11], Stuart Calder [11], Ko Mibu [12], Michel Kenzelmann [5], Liu Hao Tjeng [8], Runze Yu [1,2,4✉], Masaki Azuma [3,6✉], Changqing Jin [1,2,13] & Youwen Long [1,2,13✉]

Pb$MO_3$ ($M$ = 3$d$ transition metals) family shows systematic variations in charge distribution and intriguing physical properties due to its delicate energy balance between Pb 6$s$ and transition metal 3$d$ orbitals. However, the detailed structure and physical properties of PbFeO$_3$ remain unclear. Herein, we reveal that PbFeO$_3$ crystallizes into an unusual 2$a_p$ × 6$a_p$ × 2$a_p$ orthorhombic perovskite super unit cell with space group $Cmcm$. The distinctive crystal construction and valence distribution of Pb$^{2+}_{0.5}$Pb$^{4+}_{0.5}$FeO$_3$ lead to a long range charge ordering of the -A-B-B- type of the layers with two different oxidation states of Pb (Pb$^{2+}$ and Pb$^{4+}$) in them. A weak ferromagnetic transition with canted antiferromagnetic spins along the $a$-axis is found to occur at 600 K. In addition, decreasing the temperature causes a spin reorientation transition towards a collinear antiferromagnetic structure with spin moments along the $b$-axis near 418 K. Our theoretical investigations reveal that the peculiar charge ordering of Pb generates two Fe$^{3+}$ magnetic sublattices with competing anisotropic energies, giving rise to the spin reorientation at such a high critical temperature.

[1] Beijing National Laboratory for Condensed Matter Physics, Institute of Physics, Chinese Academy of Sciences, Beijing, China. [2] School of Physical Sciences, University of Chinese Academy of Sciences, Beijing, China. [3] Laboratory for Materials and Structures, Tokyo Institute of Technology, Yokohama, Kanagawa, Japan. [4] Tokyo Tech World Research Hub Initiative (WRHI), Institute of Innovative Research, Tokyo Institute of Technology, Yokohama, Kanagawa, Japan. [5] Laboratory for Neutron Scattering and Imaging, Paul Scherrer Institut, Villigen, Switzerland. [6] Kanagawa Institute of Industrial Science and Technology, Ebina, Japan. [7] Department of Advanced Materials and Engineering, Faculty of Engineering Sciences, Kyushu University, Kasuga, Japan. [8] Max-Planck Institute for Chemical Physics of Solids, Dresden, Germany. [9] National Synchrotron Radiation Research Center, Hsinchu, Taiwan, ROC. [10] European Synchrotron Radiation Facility, Grenoble, France. [11] Neutron Scattering Division, Oak Ridge National Laboratory, Oak Ridge, TN, USA. [12] Graduate School of Engineering, Nagoya Institute of Technology, Nagoya, Japan. [13] Songshan Lake Materials Laboratory, Dongguan, Guangdong, China. [14] These authors contributed equally: Xubin Ye, Jianfa Zhao, Hena Das. ✉email: yurz@iphy.ac.cn; mazuma@msl.titech.ac.jp; ywlong@iphy.ac.cn

Transition metal perovskite oxides (general formula: $ABO_3$) display a variety of desirable electronic and magnetic properties, such as high-temperature superconductivity[1], colossal magnetoresistance[2,3], metal–insulator transition[4], multiferroicity[5–7], and electrocatalysis[8]. $AMO_3$ ($A$ = Pb/Bi) perovskite oxides are typical examples of charge degrees of freedom at the $A$ site depending on $6s^0$ ($Pb^{4+}$, $Bi^{5+}$) and $6s^2$ ($Pb^{2+}$, $Bi^{3+}$) electron configurations for the prohibition of the $6s^1$ configuration[9–11]. For example, in Bi-based systems, $BiCrO_3$ through $BiCoO_3$ are all $Bi^{3+}M^{3+}O_3$; however, $BiNiO_3$ has an unusual valence state $Bi^{3+}_{0.5}Bi^{5+}_{0.5}Ni^{2+}O_3$ with ordered $Bi^{3+}$ and $Bi^{5+}$ charge states[9–15].

In Pb-based systems, as the $d$ level of the transition metal becomes deeper, different crystal structures and systematic charge distribution changes are observed[16–22]. Divalent lead appears in tetragonal $PbTiO_3$ and $PbVO_3$[17]. However, the disordered coexistence of $Pb^{2+}$ and $Pb^{4+}$ states (charge glass) occurs in $PbCrO_3$ ($Pb^{3+}Cr^{3+}O_3$ on average), where a simultaneous insulator-to-metal transition and a large volume collapse arise from the melting of Pb charge glass and Pb–Cr charge transfer upon pressurizing to 2.5 GPa[18]. More interestingly, a 1:3 ordered $Pb^{2+}$ and $Pb^{4+}$ and a 1:1 ordered $Co^{2+}$ and $Co^{3+}$ have been observed in $PbCoO_3$ with the charge format of $Pb^{2+}Pb^{4+}_3Co^{2+}_2Co^{3+}_2O_{12}$ ($Pb^{3.5+}Co^{2.5+}O_3$ on average)[19,20]. Moreover, a pressure-induced spin-state transition and Pb–Co intermetallic charge transfer have been discovered in this compound[19,20]. For $PbNiO_3$, the electronic configuration is $Pb^{4+}Ni^{2+}O_3$ in the presence of a single-valence $Pb^{4+}$ state[21]. Currently, in $PbMO_3$ systems, only $PbMnO_3$ and $PbFeO_3$ remain elusive. Herein, we focus on $PbFeO_3$, which has been reported by Tsuchiya et al. in 2007[22]. However, the difficulties in the synthesis of samples and in resolving the crystal structure have inhibited investigation of its structure and physical properties.

Alongside the intriguing charge properties at the Pb site, the spin degree of freedom at the $M$-site in $PbMO_3$ systems has also received significant attention. For example, a pressure-induced high-spin to low-spin-state transition of $Co^{2+}$ accompanied by an unusual increase in the resistance in $PbCoO_3$ has been reported recently[19]. The perovskite family of $RFeO_3$ ($R$ = rare earth) exhibits various spin-related properties such as multiferroicity[23–25], laser-induced ultrafast spin reorientation (SR)[26,27], and ultrafast photomagnetic recording[28], and it is anticipated that the evolution of magnetism and spin structure of $PbFeO_3$ as a function of temperature, magnetic field or pressure will be similarly rich. A characteristic feature of $RFeO_3$ is the presence of two magnetic sublattices, i.e., the $R^{3+}$ sublattice and $Fe^{3+}$ sublattice[29–32], which generate three types of competitive exchange interactions (Fe–Fe, $R$–Fe, and $R$–$R$). The strongest Fe–Fe interaction induces a canted antiferromagnetic (AFM) ordering of $Fe^{3+}$ ($S = 5/2$) spins below the Néel temperature $T_N = 600$–740 K with a Néel vector $G_x$ and a weak ferromagnetic (FM) vector $F_z$. This type of spin structure is energetically preferred for $Fe^{3+}$ moments and occurs in almost all $Fe^{3+}O_6$ perovskite-type frameworks[25,30,33–35]. Furthermore, weak $R$–$R$ interactions can result in AFM ordering in the $R$ sublattice at a low temperature (<10 K). One of the most interesting phenomena in $RFeO_3$ is the SR induced by temperature and/or magnetic field, where the alignment of $Fe^{3+}$ spin moments changes from one crystal direction to another. As the Fe spin reorientation is regarded to be closely related to anisotropic $R$–Fe magnetic exchange interactions, the current $PbFeO_3$ with nonmagnetic $A$-site Pb ions may provide a new avenue for understanding the distinct underlying mechanism of SR transition.

In this study, comprehensive investigations based on synchrotron X-ray diffraction (SXRD), neutron powder diffraction (NPD), electron diffraction (ED), hard X-ray photoemission spectroscopy (HAXPES), soft X-ray absorption spectroscopy (XAS), Mössbauer spectroscopy, characterization of magnetic and electrical properties, and density functional theory (DFT) calculations were performed to examine the structure, charge state, and magnetic properties of $PbFeO_3$. We discovered that $PbFeO_3$ crystallized into a $Cmcm$ space group with a new charge ordering of a unique -A-B-B- type, whereby in the direction of the layers stacking, one layer composed by $Pb^{2+}$ is interleaved by two layers built up from a mixture of $Pb^{4+}$ and $Pb^{2+}$ in a 3:1 ratio. Moreover, a high-temperature weak FM transition and subsequently a spin reorientation transition occurred at ~600 and 418 K, respectively. Related mechanisms are proposed to explain the SR of $PbFeO_3$.

## Results

**Crystal structure of PbFeO₃.** According to a previous report[22], the sample quality of $PbFeO_3$ is extremely sensitive to the synthesis conditions. We carefully optimized the synthesis pressure and temperature, and finally obtained a nearly single phase by synthesizing at 8 GPa and 1423 K for 30 min. The SXRD pattern of $PbFeO_3$ is shown in Fig. 1a. Except for a tiny $Fe_2O_3$ impurity phase, all the diffraction peaks can be indexed based on an unusual $2a_p \times 6a_p \times 2a_p$ orthorhombic perovskite super unit cell, where $a_p$ refers to the pseudolattice parameter of a cubic $ABO_3$ perovskite subcell. To identify the reflection conditions as well as possible space groups, we performed an ED experiment. ED patterns indicate that $PbFeO_3$ has an orthorhombic cell with a $2a_p \times 6a_p \times 2a_p$ superlattice (see Fig. 1b and Supplementary Fig. S1). The reflection conditions are $0kl$ ($k = 2n$), $h0l$ ($h$, $l = 2n$), $hk0$ ($h + k = 2n$), $h00$ ($h = 2n$), $0k0$ ($k = 2n$), and $00l$ ($l = 2n$), which are consistent with the space groups of $Cmc2_1$ (No. 36), $C2cm$ (No. 40), and $Cmcm$ (No. 63). We determined the final crystal structure using a Rietveld refinement of the SXRD pattern based on the primary structural models suggested by the ED. In comparison, we discovered that the most reliable structure model was $Cmcm$ (see Fig. 1a), as reducing the symmetry cannot improve the quality of the fit any more. Since X-rays are not sensitive to the light element of oxygen, we perform NPD to precisely determine the oxygen position. The Rietveld refinement results of NPD data collected at 300 K are illustrated in Fig. 1c. We observed a $G$-type AFM structure at 300 K, which will be discussed later. In the crystal symmetry of $Cmcm$, Pb atoms occupied six special Wyckoff positions $4c$ (0, $y$, 0.25), Fe atoms occupied two special positions $8d$ (0.25, 0.25, 0) and $16h$ ($x$, $y$, $z$), whereas O atoms occupied one $16h$ ($x$, $y$, $z$) site, three different $8f$ (0, $y$, $z$) sites, three different $8g$ ($x$, $y$, 0.25) sites, and one $8e$ ($x$, 0, 0) site. Table 1 and Supplementary Table S1 list the refined structure parameters, including the detailed lattice constants and atomic positions. The bond lengths and the results of bond valence sum (BVS) calculations for the NPD and SXRD data are listed in Table 2 and Supplementary Table S2, respectively. The obtained lattice parameters were $a = 7.89945(15)$ Å, $b = 23.46820(45)$ Å, and $c = 7.73406(15)$ Å (corresponding to perovskite cell parameters $a_p$ of 3.94973, 3.91137, and 3.86703 Å along with directions $a$, $b$, and $c$, respectively). As shown in Fig. 1d, each Fe atom is coordinated by six ligand O atoms with Fe–O distances varying from 1.95 to 2.14 Å, forming a perovskite-type $FeO_6$ octahedron framework (see Fig. 1d). Moreover, the structure refinement results demonstrate that the Pb layers perpendicular to the $b$ axis show a –shorter–longer–shorter– stacking with an interlayer distance of 3.8078 Å, 4.1186 Å, and 3.8078 Å. Figure 1e shows the high-angle annular dark-field (HAADF) image of $PbFeO_3$. It is clear that the distances for the bright spots, which are the locations of the Pb layers, show a modulation similar to the –shorter–longer–shorter– pattern mentioned above. This result further confirms the reliability of the crystal structure determined from the ED, SXRD, and NPD data.

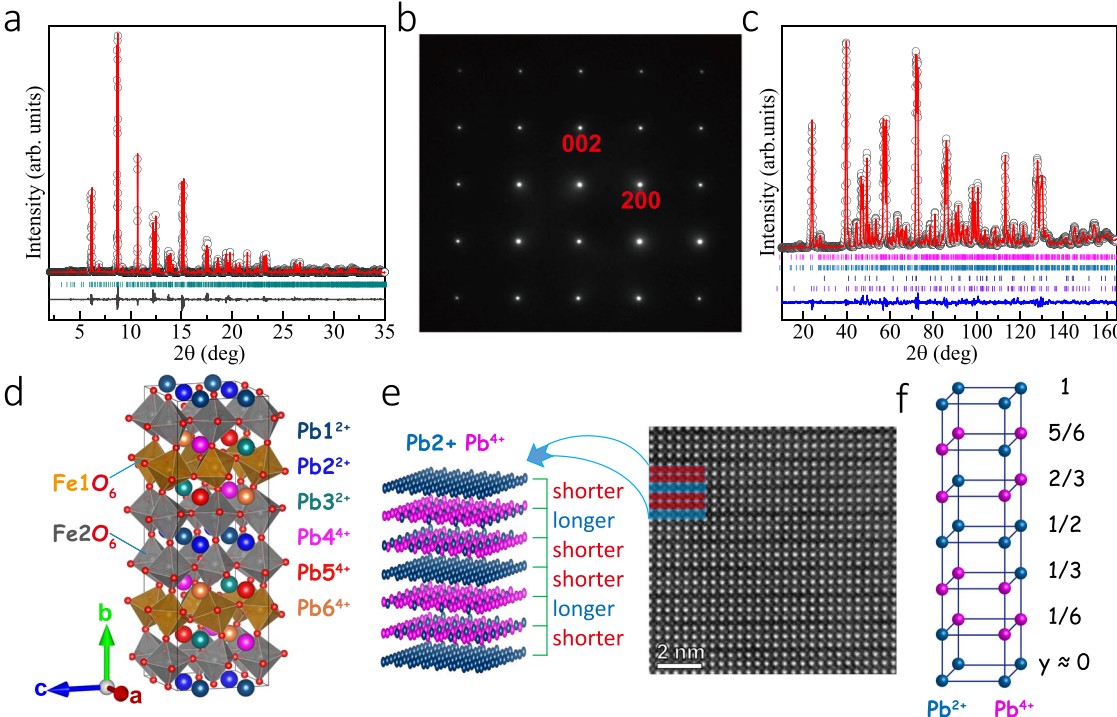

**Fig. 1 Synthesis and crystal structure characterizations of PbFeO₃. a** Rietveld refinement for SXRD pattern recorded at room temperature for PbFeO₃. Observed (black circles), calculated (red line), and difference (gray line) values are shown. The allowed Bragg positions in *Cmcm* symmetry are indicated by ticks (dark cyan). **b** ED patterns along [010] pseudocubic zone axis at RT. **c** Rieveld refinement for NPD pattern at 300 K for PbFeO₃. Observed (black circles), calculated (red line), and difference (blue) values are shown. Bragg positions of PbFeO₃ are indicated by ticks (dark cyan), which correspond to the allowed nuclear (magenta) and magnetic (dark cyan) Bragg peaks of PbFeO₃; and allowed nuclear (navy) and magnetic (violet) Bragg reflections of the impurity phase Fe₂O₃ (~5 wt%). **d** Crystal structure of PbFeO₃. **e** Left part: illustration of Pb modulations; Right part: HAADF image along [001] pseudocubic zone axis of PbFeO₃. Distances for the bright spots, which are the locations of Pb, indicate a modulation with a shorter–longer–shorter pattern. **f** Sketch of unique -A-B-B- type of charge ordering of PbFeO₃ composed of two types of differently charged layers.

**Table 1 Crystallographic parameters of PbFeO₃ refined from NPD pattern at RT[a].**

| Atom | Site | x | y | z | $100 \times B_{iso}$ (Å) |
|------|------|---|---|---|--------------------------|
| Pb1 | 4c | 0 | 0.9966 (2) | 0.25 | 1.04 (2) |
| Pb2 | 4c | 0 | 0.4999 (2) | 0.25 | 1.04 (2) |
| Pb3 | 4c | 0 | 0.8284 (2) | 0.25 | 1.04 (2) |
| Pb4 | 4c | 0 | 0.1540 (2) | 0.25 | 1.04 (2) |
| Pb5 | 4c | 0 | 0.3351 (2) | 0.25 | 1.04 (2) |
| Pb6 | 4c | 0 | 0.6585 (2) | 0.25 | 1.04 (2) |
| Fe1 | 8d | 0.25 | 0.25 | 0 | 0.52 (2) |
| Fe2 | 16h | 0.2528 (4) | 0.5841 (1) | 0.9981 (3) | 0.52 (2) |
| O1 | 8g | 0.3034 (5) | 0.7742 (2) | 0.25 | 0.941 (2) |
| O2 | 16h | 0.3094 (3) | 0.6712 (1) | 0.0549 (5) | 0.941 (2) |
| O3 | 8f | 0 | 0.2706 (2) | 0.9374 (6) | 0.941 (2) |
| O4 | 8e | 0.2781 (5) | 0 | 0 | 0.941 (2) |
| O5 | 8f | 0 | 0.0768 (2) | 0.0609 (5) | 0.941 (2) |
| O6 | 8g | 0.1769 (5) | 0.5870 (2) | 0.25 | 0.941 (2) |
| O7 | 8f | 0 | 0.6093 (2) | 0.9619 (5) | 0.941 (2) |
| O8 | 8g | 0.2781 (5) | 0.4054 (2) | 0.25 | 0.941 (2) |

[a]Space group *Cmcm* (No. 63), Z = 24. a = 7.89945 (15) Å, b = 23.46820 (45) Å, c = 7.73406 (15) Å, V = 1433.727 (4) Å³. $R_{wp}$ = 5.81%, $R_p$ = 5.27%.

**Charge-order structure of PbFeO₃.** The BVS calculations (see Table 2) reveal that the lead atoms, located at the three positions (Pb1, Pb2, and Pb3), show a valence state close to +2, but the other three (Pb4, Pb5, and Pb6) provide valence sums corresponding to a valence state close to +4. However, Fe always shows a +3 state regardless of its atomic positions. These features

indicate that the valence distribution of PbFeO₃ should be $Pb^{2+}_{0.5}Pb^{4+}_{0.5}Fe^{3+}O_3$. Based on the charge distribution in the *ac* plane with different *y*-axis values, $Pb^{2+}$ and $Pb^{4+}$ show a unique long-range charge ordering, as shown in Fig. 1f. Specifically, there exist two types of layers with respect to the estimated valence states of Pb. Within each unit cell repetition period along the crystallographic *b* axis, two identically charged layers consisting of Pb1 and Pb2 atoms characterized by a $Pb^{2+}$ charge state are located at *y* ≈ 0 and 1/2. In addition, four differently charged layers, each built up by three $Pb^{4+}$ atoms: Pb4, Pb5, and Pb6 and just one $Pb^{2+}$ charged Pb3 atom located at *y* ≈1/6, 1/3, 2/3, and 5/6, respectively. This 3:1 ratio of the $Pb^{4+}$ and $Pb^{2+}$ ions provided an average oxidation state +3.5 for Pb atoms in these layers. Therefore, a peculiar -A-B-B- charge ordering, i.e., one layer with exclusive $Pb^{2+}$ oxidation state (A) is followed by two layers (B) with an average oxidation state of $Pb^{3.5+}$, was realized in PbFeO₃. This unprecedented charge ordering renders PbFeO₃ unique in all the reported charge-order perovskite oxides[36,37], resulting in the formation of a $2a_P \times 6a_P \times 2a_P$ perovskite supercell. Neither the melting of Pb charge ordering nor the valence and spin-state changes of Fe were observed when the sample was heated to temperatures near its decomposition temperature of approximately 730 K (Supplementary Fig. S2) or cooled to a low temperature down to 10 K, as confirmed by temperature-dependent XAS (see Supplementary Fig. S3). Other charge-order compounds in $AMO_3$ perovskite systems, such as BiNiO₃, PbCrO₃, and PbCoO₃ as mentioned above[16,18,19], remained unchanged upon cooling or heating at ambient pressure; however, they exhibited pressure-induced intersite charge transfer transitions. Moreover, BiNiO₃ with R, Pb, and Sb substitution for Bi or Fe substitution

**Table 2 Pb–O and Fe–O bond lengths and BVSs[a] for PbFeO₃ refined from NPD pattern at RT.**

| | Distance (Å) | | Average bond length (Å) | BVS |
|---|---|---|---|---|
| Fe1–O | 2.090 (1) | ×2 | 2.035 | 2.82 |
| | 2.059 (1) | ×2 | | |
| | 1.955 (3) | ×2 | | |
| Fe2–O | 1.989 (2) | | 2.039 | 2.80 |
| | 2.040 (2) | | | |
| | 2.101 (2) | | | |
| | 2.137 (3) | | | |
| | 1.945 (2) | | | |
| | 2.020 (2) | | | |
| Pb1–O | 2.385 (5) | ×2 | 2.793 | 1.94 |
| | 2.766 (5) | ×2 | | |
| | 2.928 (3) | ×4 | | |
| | 2.958 (4) | ×2 | | |
| Pb2–O | 2.475 (5) | ×2 | 2.772 | 2.08 |
| | 2.610 (3) | ×4 | | |
| | 3.043 (5) | ×2 | | |
| | 3.123 (5) | ×2 | | |
| Pb3–O | 2.517 (5) | ×2 | 2.806 | 2.06 |
| | 2.713 (4) | ×2 | | |
| | 2.739 (5) | ×2 | | |
| | 2.798 (2) | ×4 | | |
| | 3.275 (3) | ×2 | | |
| Pb4–O | 2.169 (2) | ×4 | 2.577 | 3.99 |
| | 2.328 (5) | ×2 | | |
| | 2.998 (4) | ×2 | | |
| | 3.221 (4) | ×2 | | |
| Pb5–O | 2.095 (5) | ×2 | 2.451 | 3.92 |
| | 2.110 (5) | ×2 | | |
| | 2.747 (4) | ×2 | | |
| | 2.852 (5) | ×2 | | |
| Pb6–O | 2.184 (5) | ×2 | 2.535 | 3.62 |
| | 2.206 (5) | ×2 | | |
| | 2.510 (4) | ×2 | | |
| | 2.888 (2) | ×4 | | |

[a]$V_i = \sum_j S_{ij}$, $S_{ij} = \exp\{(r_0 - r_{ij})/0.37\}$. Values calculated using $r_0 = 2.112$ for $Pb^{2+}$, 2.042 for $Pb^{4+}$, and 1.751 for $Fe^{3+}$.

edge jump and the Pb 5$d$ states, the spectral intensity of the pre-edge peak (see the green curve in Fig. 2b) in PbFeO₃ is approximately half of that in PbNiO₃, indicating that the average valence state of Pb is +3. Moreover, the valence state of Pb can be further confirmed by HAXPES measurements. Figure 2c shows the HAXPES results for PbFeO₃ and other Pb$MO_3$ compounds with $M =$ Ti, Cr, Co, and Ni used as standard references. Two components appeared in both the Pb $4f_{7/2}$ and Pb $4f_{5/2}$ peaks for PbCrO₃ ($Pb^{2+}_{0.5}Pb^{4+}_{0.5}CrO_3$), PbCoO₃ ($Pb^{2+}_{0.25}Pb^{4+}_{0.75}CoO_3$), and PbFeO₃. Each peak can be deconvoluted into two Gaussians, as reported previously[18,19]. The $6s^0$ electronic configuration in Pb resulted in lower binding energy than that of $6s^2$ because of a strong screening effect[19]; hence, the components at lower binding energies are attributable to $Pb^{4+}$ ions. The peak energies of PbFeO₃ were close to those of $Pb^{2+}_{0.5}Pb^{4+}_{0.5}Cr^{3+}O_3$, indicating the coexistence of $Pb^{2+}$ and $Pb^{4+}$ ions. We estimated the fractions of $Pb^{2+}$ and $Pb^{4+}$ from the area ratios of $Pb^{2+}$ and $Pb^{4+}$ using PbCrO₃ data as the standard for $Pb^{2+}_{0.5}Pb^{4+}_{0.5}$. As shown in Fig. 2d and Supplementary Table S3, the fitting demonstrates a nearly equal ratio between $Pb^{2+}$ and $Pb^{4+}$ in PbFeO₃. Based on the BVS, XAS, and HAXPES results, we confirm that the charge configuration of PbFeO₃ is $Pb^{2+}_{0.5}Pb^{4+}_{0.5}Fe^{3+}O_3$ with novel ordered $Pb^{2+}$ and $Pb^{4+}$ distribution.

**Electrical transport property of PbFeO₃.** The electrical properties of PbFeO₃ are shown in Fig. 3a. A strong insulating feature was observed below 350 K (resistivity $\rho > 10^7$ Ω·cm), and a temperature-dependent behavior appeared with the values reduced by four to five orders of magnitude in the temperature range of 350–630 K. The data can be described by a three-dimensional (3D) variable-range-hopping (VRH) regime with the function $\rho(T) = \rho_0 \exp(T_0/T)^{1/4}$, where $T_0 = (3^3/\pi)/k_B\xi^2 D(E_F)$, $k_B$ is the Boltzmann constant, $D(E_F)$ the density of states (DOS) at the Fermi level, and $\xi$ the localization length of a wave function for localized electrons [$\exp(-r/\xi)$][50,51]. The inset of Fig. 3a shows the plot of $\ln\rho$ vs. $T^{-1/4}$ for the resistivity data of PbFeO₃, and the straight line of the plot is the fitting result obtained using the VRH model in the temperature range of 520–630 K. The fitting results yielded $T_0 = 5414$ K, and $\rho_0 = 2.0 \times 10^{-4}$ Ω·cm. This implies that in this system the electrical transport is dominated by the hopping of localized charge carriers. This feature is consistent with the intrinsic insulating behavior of the $Fe^{3+}O_6$ perovskite framework, such as LaFeO₃ and BiFeO₃, owing to the strong electronic correlation effects[47,52]. Moreover, the valence band-edge positions obtained from HAXPES indicate that the valence bond is away from the Fermi energy and the binding energy is ~1 eV (see Fig. 3b), consistent with the insulating nature of PbFeO₃. It is noteworthy that no electrical anomaly was observed at the magnetic transition temperatures as shown later.

**Electronic structures and charge-order analysis by DFT.** The crystal and electronic structures of the extraordinary charge-ordered states of $Pb^{2+}$ and $Pb^{4+}$ were further investigated by employing the DFT calculations for PbFeO₃. To gain a deeper insight, we conducted a comparative analysis of the experimentally observed charge-order structure and other typically known $A$-site charge-ordered phases. We optimized the crystal structure of various $Pb^{2+}/Pb^{4+}$ 1:1 charge-ordered phases (see Supplementary Fig. S4) considering several collinear FM and AFM arrangements of Fe spins. The corresponding results are summarized in Fig. 4. We discovered that the experimentally determined $Cmcm$ $Pb^{2+}/Pb^{4+}$ charge-order phase corresponded to the lowest energy structure and was 56 meV/f.u., which was lower compared with the nonpolar columnar-ordered $Pmmn$ structure

for Ni indicated a temperature-induced charge transfer accompanied by negative thermal expansion[15,38–43]. The investigation of current PbFeO₃ under pressure and doping effects will be performed in our future study.

The distinctive valence distribution of PbFeO₃ can be further confirmed by Fe $L_{2,3}$-edges XAS, Pb $L_3$-edge XAS, and Pb-4$f$ XPS. The sharp multiple spectral features at the 3$d$ transition element $L_{2,3}$-edges are extremely sensitive to the valence state[44,45] and local environment[46,47]. The Fe $L_{2,3}$-edge results of PbFeO₃ are shown in Fig. 2a, together with Fe₂O₃ as a $Fe^{3+}$ reference. Both of them show similar peak energy positions and spectral features, indicating the formation of the $Fe^{3+}$ ($3d^5$) state in PbFeO₃, consistent with the BVS result. Considering that the Pb $L_3$-edge spectral profile obtained from the high-resolution partial fluorescence yield (PFY) model is an effective method to identify the valence state of Pb, we performed Pb $L_3$-edge XAS using the PFY model[20,44,48]. As shown in Fig. 2b, a lower energy shoulder appeared at ~13,030 eV, which is assigned to the dipole-allowed transition from the 2$p_{3/2}$ core level to the unoccupied 6$s$ state. Typically, this pre-edge can be observed in $Pb^{4+}$ ions with two 6$s$ holes but is absent in $Pb^{2+}$ ions with fully occupied 6$s$ states[49]. Therefore, we can confirm that a $Pb^{4+}$ component exists in PbFeO₃. Meanwhile, we observed that the pre-edge peak height of PbFeO₃ was lower than that of PbNiO₃, which possesses a pure $Pb^{4+}$ state. After subtracting the background originating from the

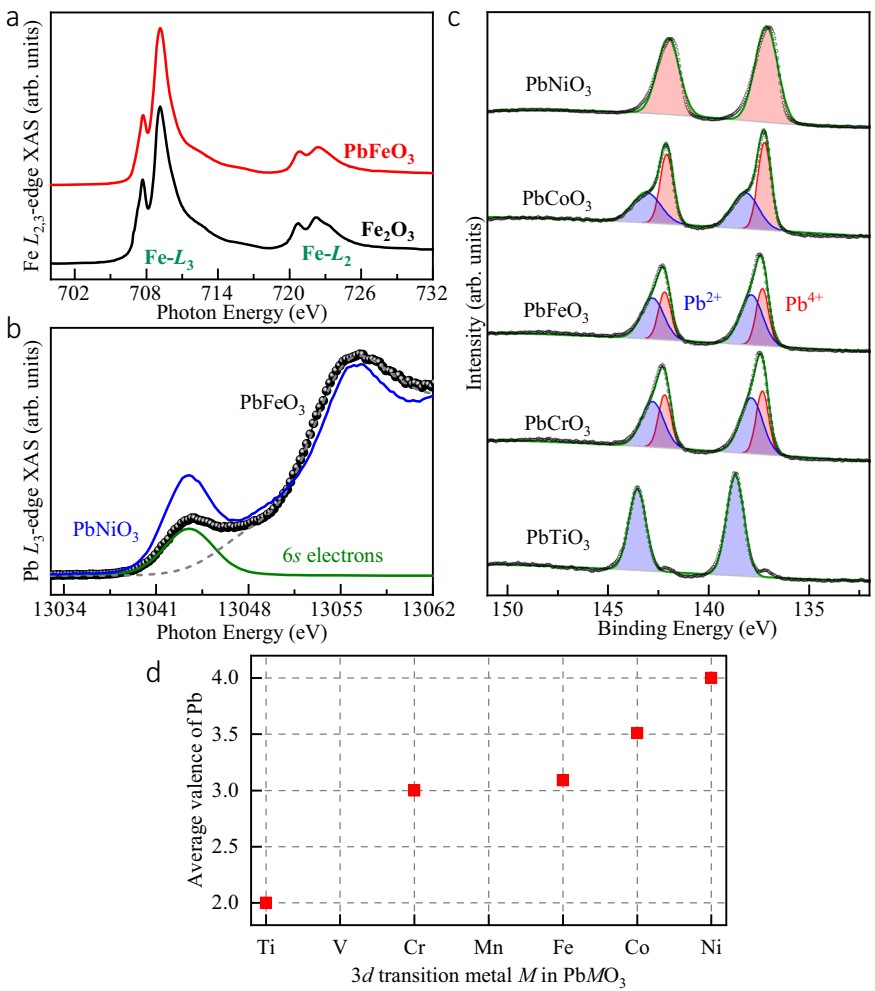

**Fig. 2 Valence-state characterization by XAS and HAXPES spectra. a** XAS of Fe $L_{2,3}$-edge of PbFeO$_3$, and reference Fe$_2$O$_3$; **b** Pb $L_3$-edge XANES of PbFeO$_3$ (black circles) and PbNiO$_3$ (blue curve) for comparison. The green curve in (**b**) shows the pre-edge peak 6$s$ state of PbFeO$_3$ after subtracting the background (dashed gray curve). **c** Pb-4$f$ HAXPES results for PbTiO$_3$, PbCrO$_3$, PbFeO$_3$, PbCoO$_3$, and PbNiO$_3$ at RT. Predominant of Pb$^{4+}$ and Pb$^{2+}$ are evident in the spectrum for PbFeO$_3$. **d** Average Pb valence state calculated from area ratios of Pb$^{2+}$ and Pb$^{4+}$ components. PbTiO$_3$, PbCrO$_3$, and PbNiO$_3$ are standard for Pb$^{2+}$, Pb$^{3+}$ (Pb$^{2+}_{0.5}$Pb$^{4+}_{0.5}$), and Pb$^{4+}$, respectively.

(see Figs. 4a and 4b). All the polar phases (*Cm*, *Pc*, and *Pm* shown in Supplementary Fig. S4), which were considered in this study, had higher energies compared with their nonpolar counterparts. The existence of two different types of lead ions in the *Cmcm* structure was reflected in our analysis of the calculated density of states (see Fig. 4c). The conduction bands within the energy range of the Fermi level to 2 eV comprised Pb 6$s$ orbitals that were strongly hybridized with O 2$p$ orbitals. This implies that within the *Cmcm* structure, half of the Pb ions were highly valent (+4), similar to several other Pb-based materials[16,53]. The other half of the Pb ions formed a 6$s^2$ electronic structure, which corresponds to the +2 valence state. Consequently, the Fe ions exhibited half-filled $t_{2g}^3 e_g^2$ electronic structures, thereby resulting in a strong AFM interaction between the Fe spins, which stabilized the *G*-type AFM structure (see Fig. 4b). The *G*-type AFM phase was insulating and indicated a bandgap of ~0.8 eV, agreeing with the experimentally determined value.

Various lattice distortions contributed to the formation of the charge-order *Cmcm* structure. As shown in Fig. 4a, the Pb charge-ordered patterns lead to the formation of layered-like components (denoted as **L1** and **L2**) made up of Fe2 ions. A layer of Fe1 ions remains sandwiched between every such **L1** and **L2** layers. We can categorize these distortions that contribute to the nonpolar *Cmcm*

structure as follows. First, based on the oxygen octahedral rotations around the crystallographic axes, we have identified, (a) in-phase oxygen octahedral rotations around the crystallographic $c$ axis, which correspond to the $M_3^+$ symmetry of the undistorted and charge disordered cubic $Pm\overline{3}m$ structure, (b) out-of-phase oxygen octahedral rotations around the $b$ axis (corresponding to the $R_4^+$ symmetry of the cubic $Pm\overline{3}m$ structure), and (c) in-phase oxygen octahedral rotations around the $a$ axis around the Fe1 ions (corresponding to the $Z4$ [1/3, 1/2, 0] symmetry of the cubic $Pm\overline{3}m$ structure), which can transform the cubic $Pm\overline{3}m$ structure to *Pmma* structure. Next, we categorize the distortions that have been observed to occur exclusively around the Fe2 ions in the layered-like components (**L1** and **L2**) of the charge-order *Cmcm* structure in response to the non-trivial arrangement of the Pb$^{2+}$ and Pb$^{4+}$ cations (as illustrated in Supplementary Fig. S9). This category of distortions can be further classified as, (a) strong anti-ferro-distortive (AFD) movements of the oxygen and Fe2 ions in response to the difference in the charge states of the PbO layers along the $b$ axis which correspond to the $DT1$ [0, 1/3, 0] symmetry (transform $Pm\overline{3}m \rightarrow P4/mmm$), (b) weak buckling in the O-Fe2-O connections along with the $a$ and $c$ axes (which follows the $DT2$ [0, 1/3, 0] symmetry transforming $Pm\overline{3}m \rightarrow Pmmm$), (c) AFD movements of the oxygens and Fe2 ions along the $a$ and $c$ axes

which transform as $Z4$ [1/6, 1/2, 0] (transform $Pm\bar{3}m \rightarrow Cmmm$, hereafter denoted as $Z4$-I) and $Z4$ [1/3, 1/2, 0] (transform $Pm\bar{3}m \rightarrow Pmma$, hereafter denoted as $Z4$-II) symmetries. The third category of distortions is the weak AFD movements of the Pb and oxygens ions following $R_5^+$, $X_5^+$, and $T3$ symmetries. The

theoretically determined Fe–O bond lengths, as listed in Fig. 4a, agree well with their experimentally determined values (see Table 2). Furthermore, similar AFD phenomena were also exhibited by other layered and layered-like charge-order structures that were taken into consideration (Supplementary Fig. S4) in this study.

**Magnetic transitions and spin structures of PbFeO₃.** The zero-field cooling (ZFC) magnetic susceptibility curve is illustrated in Fig. 5a. As the temperature decreased to $T_N \approx 600$ K, the magnetic susceptibility increased significantly, indicating a long-range magnetic transition. Upon further cooling to $T_{SR} \approx 418$ K, an abrupt decrease occurred, suggesting the presence of an SR transition. Based on the isothermal magnetization curves measured at different temperatures (see Fig. 5b), a linear magnetization behavior can be observed at 630 K, in accordance with paramagnetism. However, between $T_N$ and $T_{SR}$, considerable magnetic hysteresis occurred. For example, at 450 K, the observed coercive field was ~0.5 T. Moreover, there was a small amount of residual magnetization (0.01 $\mu_B$/Fe). These features indicate that a weak FM transition originating from the canted AFM spins would occur at $T_N$. Below 300 K, the magnetization almost resumed to a linear behavior with negligible coercive field and residual magnetization, illustrating that the AFM structure did not comprise a weak ferromagnetic component. These behaviors are similar to those observed in $R$FeO₃ perovskite oxides with a spin reorientation transition[33–35]. The Mössbauer spectrum at 300 K is shown in Supplementary Fig. S5. The spectrum can be fitted with three sets of magnetically split sextets with area ratios of 60, 34, and 6%. These components are attributed to Fe at the 16$h$ site, 8$d$ site, and at $\alpha$-Fe₂O₃ impurities. The values of the isomer shift and magnetic hyperfine field for the 16$h$ component (0.42 mm/s, 48.8 T) and those for the 8$d$ component (0.38 mm/s, 42.0 T) indicate that Fe in PbFeO₃ is trivalent and in a magnetically ordered state at RT.

To further determine the spin ordering of PbFeO₃, we performed temperature-dependent NPD measurements from 2 to 625 K, as shown in Supplementary Figs. S6, S7, and S8.

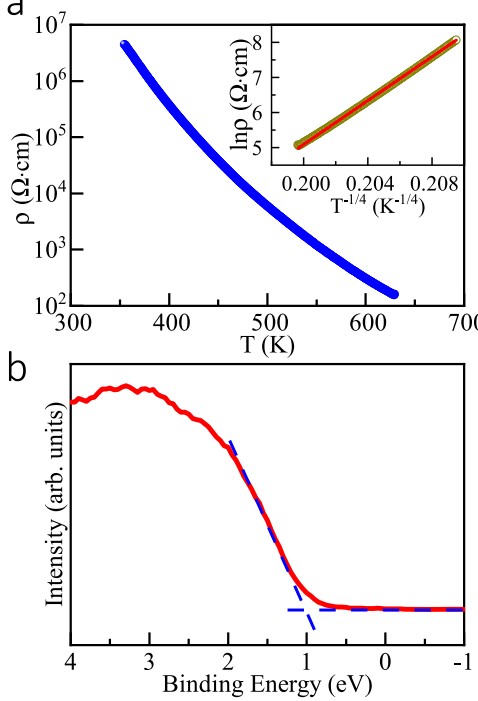

**Fig. 3 Electrical properties of PbFeO₃. a** Temperature dependence of resistivity measured above RT for PbFeO₃. Inset shows 3D VRH model fitting between 520 and 630 K. **b** HAXPES spectrum near valence band for PbFeO₃. The energy bandgap is evaluated to be ~1 eV.

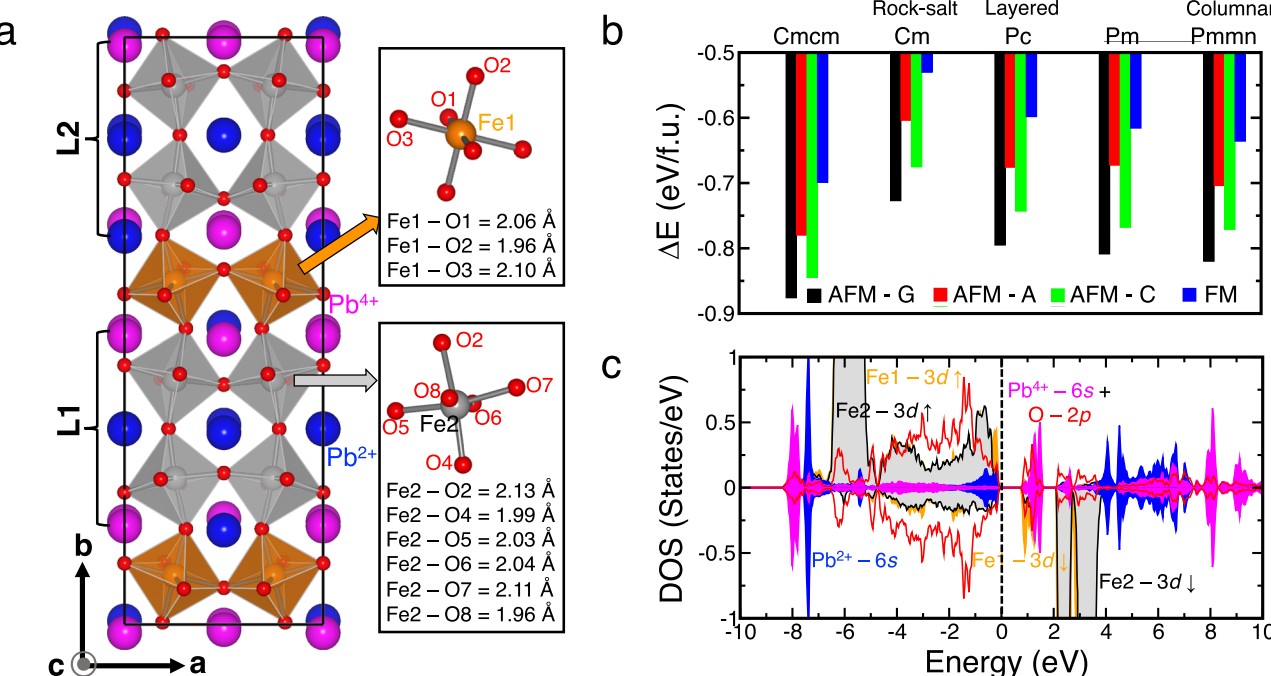

**Fig. 4 Charger-ordered phase description and electronic structure analysis. a** Description of Pb²⁺/Pb⁴⁺-ordered optimized $Cmcm$ structure. **b** Calculated energies of various Pb²⁺/Pb⁴⁺-ordered structures considering four collinear arrangements between Fe spins. Relative energy ($\Delta E$) is shown with respect to undistorted ferromagnetic phase. **c** Calculated density of states of lowest energy charge-ordered $Cmcm$ structure in $G$-type antiferromagnetic phase.

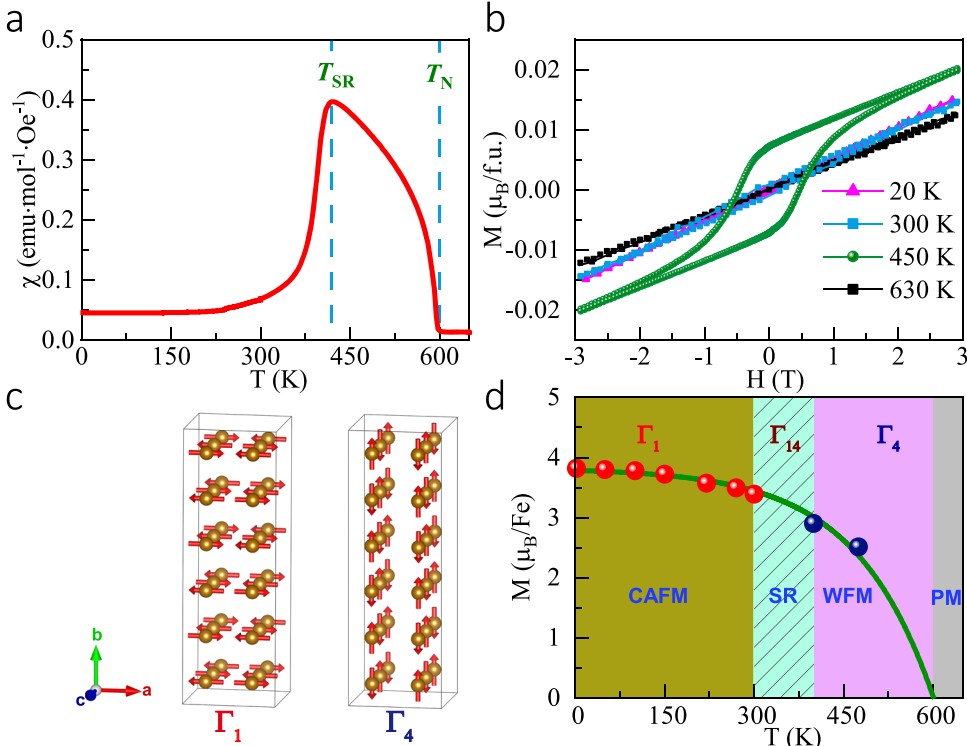

**Fig. 5 Magnetism measurements and schematic description of magnetic properties. a** Temperature dependence of magnetic susceptibility of $PbFeO_3$ measured at 0.01 T. **b** Isothermal magnetization loops measured at various temperatures. **c** Magnetic structures of $PbFeO_3$ between $T_{SR}$ and $T_N$ ($\Gamma_4$) as well as below 300 K ($\Gamma_1$). Red arrows depict $Fe^{3+}$ moments. **d** Magnetic-phase diagram for $PbFeO_3$. Red and blue spots denote magnetic moments determined from neutron diffraction. The error bars of magnetic moments are within the symbols. Green line provides visual guidance. CAFM collinear antiferromagnetism, WFM weak ferromagnetism, PM paramagnetism.

Supplementary Fig. S6 shows the diffraction patterns together with the Rietveld refinement results at 100 and 475 K. The Rietveld analysis for magnetic diffraction revealed a commensurate magnetic order with wave vector **k** = (0, 0, 0). Furthermore, a canted *G*-type AFM structure described by $\Gamma_4$ ($G_x$, $A_y$, $F_z$) (Bertaut's notation[54]) with antiferromagnetically coupled spins along the *a* axis and an allowed net magnetization along the *c* axis was determined between $T_{SR}$ and $T_N$. The weak FM component had likely contributed to the magnetic hysteresis observed in the M–H measurement at 450 K; however, it was extremely small to be determined by the refinement. When the temperature was below $T_{SR}$, a collinear *G*-type AFM structure described by $\Gamma_1$ ($A_x$, $G_y$, $C_z$) with spin along the *b* axis was also confirmed by the refinement, consistent with the linear feature of the magnetization shown in Fig. 5b. It is apparent that the symmetry of $\Gamma_1$ ($A_x$, $G_y$, $C_z$) forbids a FM component, which is also consistent with the absence of magnetic hysteresis below $T_{SR}$. Supplementary Fig. S7 shows the NPD fitting for several characteristic peaks with the spin direction along *a* axis ($\Gamma_4$), *b* axis ($\Gamma_1$), and *c* axis ($\Gamma_2$) for comparison. It is evident that the $\Gamma_4$ and $\Gamma_1$ spin models can provide the most reliable fitting for the higher and lower temperature magnetic phases, respectively. Therefore, our studies confirmed a temperature-induced SR from $\Gamma_4$ ($G_x$, $A_y$, $F_z$) to $\Gamma_1$ ($A_x$, $G_y$, $C_z$) in $PbFeO_3$. Figure 5c shows the two different types of spin configurations. The refined magnetic moments at $T = 475$ and 100 K in units of $\mu_B$ were $M_a = 2.57$ (2) and $M_b = 3.78$ (1), respectively. Figure 5d shows the magnetic-phase diagram of $PbFeO_3$ based on the current results. With decreasing temperature, the compound changed from paramagnetic to canted AFM with a small amount of net FM moment at the critical temperature of $T_N \approx 600$ K. Upon further cooling to $T_{SR} \approx 418$ K, a continuous SR transition occurred, changing the weak FM

phase to a collinear AFM phase. The evaluated temperature range for the SR was between 300 and 418 K. Temperature-dependent magnetic moments for various phases are also shown in the phase diagram. Based on the refinement, the magnetic moment at $T = 2$ K in units of $\mu_B$ was $M_b = 3.8$ (1), which was somewhat lower than the expected value (5 $\mu_B$) for an $Fe^{3+}$ ion probably due to the considerable covalence/hybridization effects between Fe and O atoms (see Fig. 5c).

**Possible origins of spin reorientation in $PbFeO_3$.** In Fe-based perovskite oxides $RFeO_3$[33,34,55], the anisotropic *R*–Fe magnetic interactions are generally identified as driving factors behind the SR transitions. In the $PbFeO_3$ system under investigation, the non-magnetic nature of the *A*-site $Pb^{2+}$ and $Pb^{4+}$ ions negates the possibility of magnetic interactions between *A*- and *B*-site cations. However, $PbFeO_3$ still exhibits a SR transition at a much higher critical temperature than those observed in most $RFeO_3$ perovskites. This indicates that distinct mechanisms of SR transition are operational in $PbFeO_3$. We further explored this phenomenon by the application of first-principles electronic structure calculations and finite temperature Monte Carlo simulations. Our investigations are based on the spin Hamiltonian as described below Eq. (1),

$$\mathcal{H} = \sum_{n,n'} J_{nn'}^{11} \mathbf{S}_n^1 \cdot \mathbf{S}_{n'}^1 + \sum_{n,m} J_{nm}^{12} \mathbf{S}_n^1 \cdot \mathbf{S}_m^2 + \sum_{m,m'} J_{mm'}^{22} \mathbf{S}_m^2 \cdot \mathbf{S}_{m'}^2$$
$$+ \sum_n [D^1(\mathbf{S}_{nz}^1)^2 + E^1\{(\mathbf{S}_{nx}^1)^2 - (\mathbf{S}_{ny}^1)^2\}]$$
$$+ \sum_m [D^2(\mathbf{S}_{mz}^2)^2 + E^2\{(\mathbf{S}_{mx}^2)^2 - (\mathbf{S}_{my}^2)^2\}]$$

$$(1)$$

Here, the first, second, and third terms denote the symmetric exchange interactions between individual Fe spins ($\mathbf{S}^1$ and $\mathbf{S}^2$ denote

Fe1 and Fe2 spin, respectively). The longitudinal and transverse anisotropy for the Fe1 and Fe2 ions are represented as ($D^1$ and $E^1$) and ($D^2$ and $E^2$), respectively. The magnetic interaction strengths estimated on the basis of the crystal structure experimentally determined at 300 K (denoted as $S_{Expt}$) and the first-principles optimized structure (denoted as $S_{Opt}$) are listed in Supplementary Fig. S10. The interactions between Fe spins, corresponding to either of these structures, are both strong and antiferromagnetic in nature leading to G-type AFM order. We also found that, while the Fe1 ions energetically favor the orientation of the spins along the b axis, the Fe2 spins, in contrast, favor the a-axis spin orientation. Moreover, the magnetic anisotropy of the Fe1 magnetic sublattice is stronger than that of the Fe2 sublattice. Our results of total energy calculations identified the $G_y$ phase with spins parallel to the b axis to be the lowest energy phase for both the crystal structures, with energy of ~0.020 meV/Fe lower than that of the $G_x$ phase with spins along the a axis. This is consistent with experimental observations. The Monte Carlo simulated magnetic transition temperature is around ~582 K (see Supplementary Fig. S11), which is also in good agreement with the experimentally observed $T_N$. Below 582 K, both Fe1 and Fe2 sublattices exhibit G-type AFM ordering with their spins oriented along the b axis (see Supplementary Fig. S11). The spin Hamiltonian premised on $S_{Opt}$ exhibits similar behavior (see Supplementary Fig. S11). Here, the anti-symmetric and anisotropic–symmetric exchange interactions between the Fe spins that lead to the spin canting were not taken into consideration, as these interactions, being weak for $Fe^{3+}$ ions, are least likely to cause a spin orientation transition within a temperature interval of ~180 K.

The peculiar arrangement of the Pb ions leads to the creation of the two magnetic Fe1 and Fe2 sublattices with mutually competing magnetic anisotropic energies. We, therefore, explored the possible origin of SR transition through the nature of modulation of single-ion anisotropic energies. We calculated the total energies of the magnetic phases ($G_x$, $G_y$, and $G_z$) as functions of the modulation of the structural distortions mentioned earlier. Among the various distortions in $S_{Expt}$, only the symmetric distortions which owe their origin to the special arrangements of Pb cations, and eventually are transformed following $DT1$, $DT2$, and $Z4$ symmetries (see Supplementary Fig. S9), were taken into consideration. Our results (as shown in Fig. 6) indicate that an increase in the $DT1$ distortion results in the relative shortening of the average Fe1–O bond length along the b axis, which in turn enhances the magnetic anisotropy energies of Fe1 ions, contributing to the enhancement of stability of the $G_y$ phase. This stability of the $G_y$ phase is maintained within a range of ~0.8 Å modulation of the amplitude of this distortion with respect to $S_{Expt}$. An increase in the $DT2$ and $Z4$-I distortions, which modulates the Fe2–O–Fe2 bond lengths and angles in the ac plane, also increases the stability of the $G_y$ phase. The decrease in $DT2$ and $Z4$-I distortions, however, gives rise to the magnetic-phase transition from $G_y$ to $G_x$ phase (as shown in Fig. 6b, c). The same phase transition (i.e., from $G_y$ to $G_x$ phase) is also observed by increasing the AFD that follows the $Z4$-II symmetry along the c axis (see Fig. 6d). Interestingly, a decrease in the amplitude of this distortion below ~0.4 Å indicates a transition from the $G_y$ to $G_z$ phase. Our results are summarized as follows. (1) The $G_z$ phase, in most cases, is higher in energy in comparison to the $G_x$ and $G_y$ phases. (2) $DT2$ and $Z4$-I are the two distortions that have been identified to contribute weakly to the formation of the Cmcm structure and significantly in the process of magnetic-phase transitions. Notably, the transformation of these distortions from zero to any finite value has no bearing on the overall symmetry of the structure. (3) Modulation of these distortions with respect to temperature can result in the transition from $G_x$ to $G_y$ magnetic phase. Notably, there is an exact match of the results

between total energy calculations and Monte Carlo simulations based on estimated parameters of the Spin Hamiltonian as a function of $DT2$ and $Z4$-I structural distortions (as presented in Supplementary Fig. S12).

## Discussion

In summary, the crystal structure and physical properties of $PbFeO_3$ were investigated in detail through SXRD, ED, XAS, HAXPES, NPD, and theoretical calculations. It was discovered that $PbFeO_3$ possessed an unusual $2a_p \times 6a_p \times 2a_p$ orthorhombic superlattice with space group Cmcm. The BVS, XAS, HAXPES, and DFT calculation results revealed that the electronic configuration was $Pb^{2+}_{0.5}Pb^{4+}_{0.5}Fe^{3+}O_3$. A unique long-range charge ordering of the -A–B–B-type of the layers of Pb atoms with different oxidation states occurred. In the direction of the layers' stacking, one layer with exclusively $Pb^{2+}$ oxidation state (A) was followed by the two layers (B) with an average oxidation state of $Pb^{3.5+}$, which in turn was a result of mixing $3 \times Pb^{4+}$ and $1 \times Pb^{2+}$ atoms in them. The single $Pb^{2+}$ state formed two identically charge layers (A) at $y \approx 0$ and 0.5, whereas the layers comprised of 3:1 ordered $Pb^{4+}$ and $Pb^{2+}$ atoms contributed to four other charge layers at the positions y ≈1/6, 1/3, 2/3, and 5/6. When the temperature decreased to 600 K, $PbFeO_3$ experienced a canted AFM phase transition with a weak net FM moment. Further cooling induced a spin reorientation transition at ~418 K, thereby changing the canted AFM structure to a collinear one; the magnetic Fe moments were aligned along the a and b axes above and below the SR temperatures, respectively. The $PbFeO_3$ system under investigation exhibits a unique phenomenon where two magnetic Fe1 and Fe2 sublattices are created as a result of the peculiar arrangement of $Pb^{2+}$ and $Pb^{4+}$ ions. The mutually competing magnetic anisotropic energies of these two sublattices is a plausible contributing factor for the spin reorientation in $PbFeO_3$ at a higher critical temperature of 418 K. This is different from the case of $RFeO_3$ perovskites where the presence of a magnetic rare-earth ion plays the most important role in the observed SR transitions. Since the magnetic sublattices owe their origin to a certain arrangement of $Pb^{2+}/Pb^{4+}$ ions, this may introduce a unique opportunity of inducing magnetic phase transition ($M = 0 \leftrightarrow M \neq 0$, where $M$ denotes net magnetization in the system) by driving a redistribution of Pb ions via an external electric field and/or strain. This work provides a new avenue for studying the charge ordering phase and distinctive spin orientation transition with potential applications in advanced spintronic devices due to the high transition temperature and possible tuning.

## Methods

**Materials synthesis**. Polycrystalline samples of $PbFeO_3$ were prepared using a high-pressure and high-temperature method. Starting materials of high purity (>99.9%) PbO, $PbO_2$, and $Fe_2O_3$ were thoroughly mixed in a 1:1:1 molar ratio within an argon-filled glovebox, and then sealed into gold capsules of diameter 3 mm and height 3.5 mm. The capsule was treated at 8 GPa and 1423 K for 30 min in a cubic-anvil-type high-pressure apparatus and quenched to room temperature prior to the slow release of pressure. The temperature window was narrow. Many impurities were introduced when the temperature increased or decreased by 50 K.

**Characterizations**. The SXRD data for $PbFeO_3$ were collected using a large Debye–Scherrer camera installed at the BL02B2 beamlines of SPring-8 with wavelengths of 0.41965 Å. The structure was refined using the Rietveld method via the GSAS program[56]. Temperature-dependent NPD at high temperatures (>300 K) was measured at HB-2A at the High Flux Isotope Reactor of the Oak Ridge National Laboratory[57] with a wavelength of 2.41 Å. NPD patterns were obtained at ≤300 K using a high-resolution diffractometer HRPT[58] at the Swiss Spallation Neutron Source of the Paul Scherrer Institute with a wavelength of 1.886 Å, and a vanadium container measuring 6 mm in diameter was used. The NPD data were analyzed using the Rietveld package, Fullprof[59]. The ED patterns and high-angle annular dark-field (HAADF) images at room temperature (RT) were obtained using a JEOL JEM-ARM200F (scanning) transmission electron microscope. The field dependence of the isothermal magnetization ($M$) and temperature-dependent

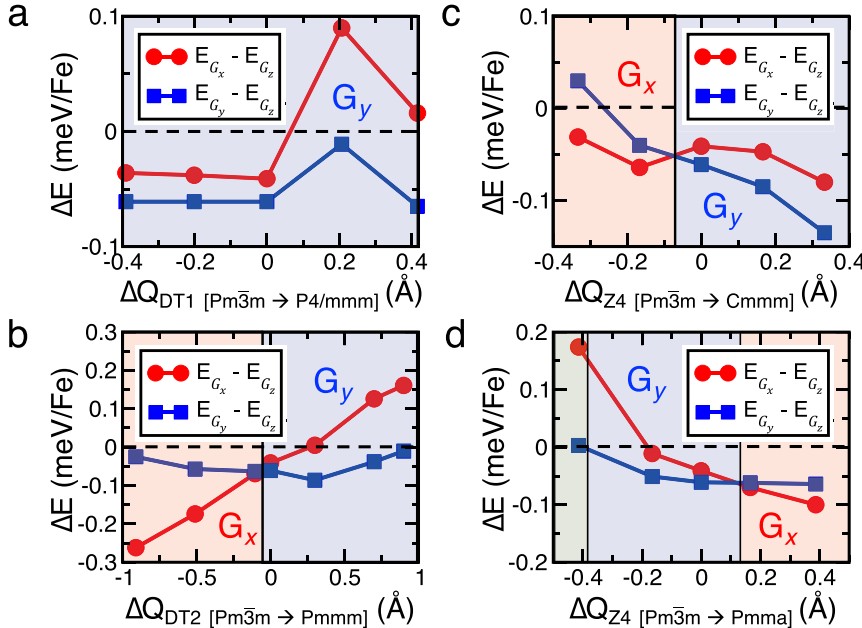

**Fig. 6 Effects of structural modulations on the stability of magnetic phases of PbFeO₃.** Calculated relative energies of $G_x$, $G_y$, and $G_z$ magnetic phases as functions of the modulation of the structural distortions corresponding to the $DT1$ (transforms $Pm\bar{3}m \rightarrow P4/mmm$, **a**), $DT2$ (transforms $Pm\bar{3}m \rightarrow Pmmm$, **b**), $Z4$ (transforms $Pm\bar{3}m \rightarrow Cmmm$, **c**) and $Z4$ (transforms $Pm\bar{3}m \rightarrow Pmma$, **d**) symmetries with respect to the crystal structure experimentally determined at 300 K ($S_{\mathrm{Expt}}$).

magnetic susceptibility ($\chi$) below 400 K were measured using a Quantum Design superconducting quantum interference device magnetometer. High-temperature magnetic susceptibility data at 400–650 K were collected using a MicroSense vibrating sample magnetometer. The resistivity at high temperatures was measured using the standard four-probe method. The size of the sample was ~2 × 1 × 1 mm for the electrical measurements. The valence states of PbFeO₃ were determined via soft X-ray absorption spectroscopy and hard X-ray absorption near-edge spectroscopy (XANES). The XAS of Fe-$L_{2,3}$ was collected at beamline BL11A of the National Synchrotron Radiation Research Center in Taiwan using the total-electron-yield mode. The spectrum of the single-crystal Fe₂O₃ sample was measured simultaneously to serve as an absolute energy reference. The high-resolution partial fluorescence yield Pb-$L_3$ XAS spectra were measured at Pb $L_1$ emission line with an overall resolution of ~2 eV at beamline ID 20 of the Synchrotron SOLEIL, France. HAXPES measurements for Pb $4f$ core levels and valence bands were performed at RT with $E = 7930.1$ eV to investigate the valence state using a hemispherical photoelectron analyzer (R4000, VG Scienta) installed at BL09XU of SPring-8. Powder samples of Pb$M$O₃ with $M$ = Ti, Cr, Fe, and Ni were pasted onto a carbon tape. Carbon black powder was mixed with PbTiO₃ before it was pasted onto a carbon tape to prevent charge-up due to the high insulation of the sample. The binding energy was calibrated using the Fermi edge of a gold film sample. Mössbauer spectroscopy was performed on a ⁵⁷Fe-25%-enriched sample at RT using the conventional absorption method.

**First-principles calculations.** The crystal and electronic structures of various Pb²⁺/Pb⁴⁺ charge-ordered states were studied by employing the DFT+$U$[60] approach using the projector-augmented plane-wave method[61] as implemented in the VASP code[62,63]. The Perdew–Burke–Ernzerhof (PBE)[64] exchange-correlation functional was used. We constructed the initial structure of various $A$-site charge-ordered phases by freezing the anti-ferrodistortive displacements of oxygen ions and the oxygen octahedra rotations (which consequently created an $A$-site oxygen polyhedral with variation in size and oxygen coordination number) within the cubic $Pm\bar{3}m$ structure, which comprised eight chemical formula unit cell sizes. The corresponding structures were optimized considering several collinear arrangements of the Fe spins. We set the screened Coulomb interaction ($U$) to 4.5 eV and 1.0 eV for Hund's coupling ($J_H$) on the Fe site. A kinetic-energy cutoff value of 500 eV and a Hellman–Feynman force convergence criterion of 0.01 eV Å⁻¹ were used. A k-point mesh was considered based on the crystal symmetry. We further cross-checked the electronic structures of the charge-ordered states using the linearized augmented plane-wave (LAPW) method as implemented in the Wien2k code[65] using the same values of $U$ and $J_H$ at the Fe site and k-point mesh, as used in the VASP code. We used the −7.5 Ry energy cutoff to separate the valence states from the core states. We calculated values of the symmetric exchange interactions between Fe spins and single-ion magnetic anisotropy energies of Fe1 ($E^1$ and $D^1$) and Fe2 ($E^2$ and $D^2$) ions by employing the linearized augmented plane-wave

(LAPW) method. We estimated SIA parameters associated to Fe1 and Fe2 magnetic sublattices by modulating the lattice distortions which owe their origin to the special Pb²⁺/Pb⁴⁺ ordered pattern. The Monte Carlo simulations based on the first-principles parameterized spin Hamiltonian were conducted considering 8 × 4 × 8 (6144 magnetic ions) supercell size of the $Cmcm$ structure and maximum sample number of 10⁹.

## Data availability
The data that support the findings of this study are available from the corresponding authors upon reasonable request.

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

## Acknowledgements

This work was supported by the National Natural Science Foundation of China (grant nos. 11934017, 51772324, 11921004, 11904392), the National Key R&D Program of China (grant nos. 2018YFE0103200, 2018YFA0305700), the Beijing Nature Science Foundation (grant nos. Z200007, 2202059), the Chinese Academy of Sciences (grant nos. XDB33010200, QYZDB–SSW–SLH013), and Grants-in-Aid for Scientific Research, JP18H05208, JP19K05246, and JP19H05625 from the Japan Society for the Promotion of Science (JSPS). The theoretical investigation was supported by the Grants-in-Aid for Scientific Research 19K05246 from the JSPS and TSUBAME supercomputing facility. The synchrotron radiation experiments were performed at SPring-8 with the approval of the Japan Synchrotron Radiation Research Institute (2018A1667, 2018B1636, and 2019B1896). This work is partly based on experiments carried out at the Swiss Spallation Neutron Source SINQ (grant no. 200021_165855), Paul Scherrer Institute, Villigen, Switzerland. We acknowledge the support from the Max Planck-POSTECH-Hsinchu Center for Complex Phase Materials. This research used resources at the High Flux Isotope Reactor, a DOE Office of Science User Facility operated by ORNL. The Möss-bauer spectroscopic measurement was performed at Nagoya Institute of Technology under the Nanotechnology Platform Program of MEXT, Japan.

## Author contributions

Y.W.L., R.Y., and M.A. conceived the study; X.Y., J.Z., L.C., and C.J. prepared the sample, X.Y., Z.L., and L.Z. performed resistance measurements; D.S., J.Y., and M.K. measured LT neutron diffraction; X.W., Z.H., H.L., C.T.C., C.S., A.E., and L.T. measured XES and XAS; Y.S., T.N., and M.A. measured SXRD and HAXPES; H.C. and S.C. measured HT-NPD. K.M. measured Mössbauer spectrum. S.W. and H.D. performed DFT calculations. C.D., H.H., X.Y., and J.Z. analyzed crystal structure. H.H. performed ED measurements. All the authors discussed the results; Y.W.L., X.Y., R.Y., D.S., H.D., and M.A. wrote the paper with comments from all the other authors.

## Competing interests

The authors declare no competing interests.
