## [Peer Review File · Nature Communications]

Reviewers' Comments:

Reviewer #1:

Remarks to the Author:

The work describes the synthesis and characterisation of perovskite PbFeO_3 , which among 3d PbMO_3 materials had remained elusive. The manuscript shows generally good scholarship, and the material presents intriguing properties that may be of interest to a wide range of researchers.

However, the discussion is very thin and offers no particularly novel insight. The manuscript boils down to a set of experimental observations. While these are undeniably important, I believe they fall short of the standard required for publication in Nature Communications.

Another minor comment

p.8 the description of the charge ordering is repeated twice in the same paragraph.

Reviewer #2:

Remarks to the Author:

The manuscript by Ye et al. reports a comprehensive structural study of PbFeO_3 perovskite, revealing a charge disproportionation of Pb and very complex charge ordering in the A-site perovskite position. In spite of the fact that I fully appreciate the complexity and exotic character of the charge ordering, I do not think the work opens a new paradigm in the field of perovskite materials. The phenomenon of charge disproportionation and the associated A-site charge ordering are well known in Pb and Bi based perovskites. The fact that the charge ordering pattern is more complicated than in other perovskites does not make it a groundbreaking discovery. Determination and refinement of such complex superstructure is far from to be trivial and from the crystallographic point of view this is an excellent work. I however do not see a broad impact of this research and in my opinion crystallographic or a journal specializing in structural chemistry will be more appropriate.

Regarding the observed spin reorientation transition, I also do not think it represents an evidence for new physics. The crystal structure is heavily distorted, implying a very complex anisotropic antisymmetric exchange consisting of several terms (associated with different distortion modes). These terms might compete with each other as well as with the single ion anisotropy (which is not negligible in many cases, in spite of the nominally isotropic electronic configuration of Fe^{3+}), giving rise to the spin reorientation.

Reviewer #3:

Remarks to the Author:

The paper by Ye et al entitled "Observation of novel charge ordering and spin reorientation in perovskite oxide PbFeO_3 " reports on an in-depth study of the structure and electronic properties of PbFeO_3 , a unique compound belonging to an intriguing family of perovskite oxides. The authors skillfully use a wide array of experimental and first-principles approaches to characterize the structure, the charge ordered state and the spin configuration of PbFeO_3 . They find a one of a kind superlattice that hosts a charge ordered arrangement of Pb^{2+} and Pb^{4+} cations. Furthermore, they characterize the magnetic spin ordering as a function of temperature and identify a spin reorientation transition between a high temperature state with canted antiferromagnetic spins (with some ferromagnetism) and a colinear AFM state at low temperature.

The paper is very clearly written and well motivated. The study relied on a wide array of state-of-the-art experimental tools, including SXRD, electron diffraction, XAS, HAXPES and NPD. The theoretical calculations have also been performed skillfully and nicely complement the

experimental analysis. The science is very interesting and should appeal to a broad cross-section of physics and materials science. I recommend publication.

Response to the Reviewers for NCOMMS-20-27977

First of all, we would like to appreciate the three reviewers very much for their valuable comments and suggestions on our manuscript with the code NCOMMS-20-27977. On the basis of these comments, we revised the manuscript. All the revisions are shown in red in the revised paper. The following is our point-by-point response to the referees.

Response to Reviewer #1

General comment: The work describes the synthesis and characterisation of perovskite PbFeO_3 , which among $3d$ PbMO_3 materials had remained elusive. The manuscript shows generally good scholarship, and the material presents intriguing properties that may be of interest to a wide range of researchers. However, the discussion is very thin and offers no particularly novel insight. The manuscript boils down to a set of experimental observations. While these are undeniably important, I believe they fall short of the standard required for publication in Nature Communications.

General response: We thank the referee very much for positive comments that the manuscript is good scholarship and intriguing. Following the referee's suggestion, we performed additional theoretical investigations based on Monte Carlo simulations and DFT calculations to find a link between the unique charge ordering and observed spin reorientation (SR) of PbFeO_3 . The new theoretical results are illustrated in Fig. 6 in the main text as well as in Figs. S5 and S10-12 in the supporting information, and detailed description and discussion are shown in red on pages 15-18 in the revised manuscript. Briefly speaking, the peculiar charge-order pattern of Pb^{2+} and Pb^{4+} ions generates two magnetic Fe1 and Fe2 sublattices. The mutually competing magnetic anisotropic energies between these two sublattices can contribute to the spin reorientation in PbFeO_3 at a higher critical temperature of 418 K. This is completely different from the case of RFeO_3 perovskites where the presence of a magnetic rare earth (R) ion plays the most important role for the SR transitions via the anisotropic R-Fe interactions.

Therefore, a new SR mechanism related with the unique charge ordering was clarified in the current PbFeO_3 . This work provides a new avenue for studying the novel charge ordering phase and distinctive spin orientation transition with potential applications in advanced spintronic devices due to the high transition temperature and possible tuning.

Detailed Comment 1: p.8 the description of the charge ordering is repeated twice in the same paragraph.

Detailed Response 1: Thanks for this point. We have revised the description to avoid any repetition as: “Therefore, a peculiar -A-B-B- charge ordering, i.e., one layer with exclusive Pb^{2+} oxidation state (A) is followed by two layers (B) with an average oxidation state of $\text{Pb}^{3.5+}$, was realized in PbFeO_3 ”. For details, please see the red on page 8 in the revised manuscript.

Response to Reviewer #2

Comment 1: The manuscript by Ye et al. reports a comprehensive structural study of PbFeO_3 perovskite, revealing a charge disproportionation of Pb and very complex charge ordering in the A-site perovskite position. In spite of the fact that I fully appreciate the complexity and exotic character of the charge ordering, I do not think the work opens a new paradigm in the field of perovskite materials. The phenomenon of charge disproportionation and the associated A-site charge ordering are well known in Pb and Bi based perovskites. The fact that the charge ordering pattern is more complicated than in other perovskites does not make it a groundbreaking discovery. Determination and refinement of such complex superstructure is far from to be trivial and from the crystallographic point of view this is an excellent work. I however do not see a broad impact of this research and in my opinion crystallographic or a journal specializing in structural chemistry will be more appropriate.

Response 1: We thank the referee for his/her comments on our manuscript. As is well known, charge order is a broad interesting topic in condensed matter physics and material sciences. The variation of charge-order pattern can induce a series of sharp and joint transformations in crystal structure and physical properties. In this work, we report the first finding of a new –A-B-B-type charge ordering. The referee mentioned that “the phenomenon of charge disproportionation and the associated A-site charge ordering are well known in Pb and Bi based perovskites”. However, the presently observed charge-order pattern has never been found to occur in any other perovskite systems. Moreover, as described in the revised manuscript (see the red on pages 15-18, Fig. 6, and Figs. S5 and S10-12), the peculiar arrangement of Pb^{2+} and Pb^{4+} ions generates two magnetic Fe1 and Fe2 sublattices. In sharp contrast to the anisotropic R-Fe magnetic interactions in RFeO_3 perovskites (R = rare earth), the mutually competing magnetic anisotropic energies of these two Fe sublattices in PbFeO_3 can trigger a spin reorientation at a higher critical temperature of 418 K. Therefore, the new charge ordering contributes to novel physical mechanism for spin reorientation. We thus believe that these new findings reported in our manuscript represent significant advances in condensed matter physics and are suitable for publication in Nat. Commun. according to this journal’s criteria as: “Nature Communications is an open access journal that publishes high-quality research from all areas of the natural sciences. Papers published by the journal represent important advances of significance to specialists within each field” (also see: <https://www.nature.com/ncomms/about>).

Comment 2: Regarding the observed spin reorientation transition, I also do not think it represents an evidence for new physics. The crystal structure is heavily distorted, implying a very complex anisotropic antisymmetric exchange consisting of several terms (associated with different distortion modes). These terms might compete with each other as well as with the single ion anisotropy (which is not negligible in many cases, in spite of the nominally isotropic electronic configuration of Fe^{3+}), giving rise to the spin reorientation.

Response 2: Thanks for these comments. To get a deeper insight for the origin of the spin reorientation observed in PbFeO_3 , we analyzed various structure distortions and carried out additional theoretical investigations based on Monte Carlo simulations and DFT calculations. The new analysis and theoretical results are shown in Fig. 6 in the main text as well as in Figs. S5 and S10-12 in the supporting information, and detailed description and discussion are shown in red on pages 12-13 and 15-18 in the revised manuscript. Briefly speaking, we find that the peculiar charge-order pattern of Pb^{2+} and Pb^{4+} ions generates two magnetic Fe1 and Fe2 sublattices. The mutually competing magnetic anisotropic energies of these two sublattices give rise to the spin reorientation in PbFeO_3 at a higher critical temperature of 418 K. This origin is completely different from the case of $R\text{FeO}_3$ perovskites where the presence of a magnetic rare earth (R) ion plays the most important role for the SR transitions via the anisotropic R-Fe interactions. In the present study of PbFeO_3 , we propose that, the nature of modulation of the anisotropic energies of Fe1 and Fe2 ions as a function of structural distortion contributes to the SR transition. Additionally, anti-symmetric and anisotropic-symmetric exchange interactions between the Fe spins can also lead to the same effect. Therefore, a new SR mechanism related with the unique charge ordering is revealed in the current PbFeO_3 .

Response to Reviewer #3

General comment: The paper by Ye et al entitled “Observation of novel charge ordering and spin reorientation in perovskite oxide PbFeO_3 ” reports on an in-depth study of the structure and electronic properties of PbFeO_3 , a unique compound belonging to an intriguing family of perovskite oxides. The authors skillfully use a wide array of experimental and first-principles approaches to characterize the structure, the charge ordered state and the spin configuration of PbFeO_3 . They find a one of a kind superlattice that hosts a charge ordered arrangement of Pb^{2+} and Pb^{4+} cations.

Furthermore, they characterize the magnetic spin ordering as a function of temperature and identify a spin reorientation transition between a high temperature state with canted antiferromagnetic spins (with some ferromagnetism) and a colinear AFM state at low temperature. The paper is very clearly written and well motivated. The study relied on a wide array of state-of-the-art experimental tools, including SXRD, electron diffraction, XAS, HAXPES and NPD. The theoretical calculations have also been performed skillfully and nicely complement the experimental analysis. The science is very interesting and should appeal to a broad cross-section of physics and materials science. I recommend publication.

General response: We appreciate the referee very much for his/her positive comments, that the manuscript is very interesting and should appeal to a broad cross-section of physics and materials science, and recommendation for publication in *Nature Communications*.

Reviewers' Comments:

Reviewer #1:

Remarks to the Author:

I appreciate the work the authors have done to improve the manuscript. The addition of calculations provide interesting insights regarding the magnetism. However, I still do not believe that these results meet the standard required for publication in Nature Communications. The observation of complicated charge ordering is certainly interesting, but better suited to a more specialised journal.

Reviewer #2:

Remarks to the Author:

The new theoretical insight into the mechanism of the spin-reorientation transition and its discussion based on the symmetry adapted distortion modes significantly improves the manuscript and I am happy to recommend the revised version for publication in Nature Communication.

Response to the Reviewers for NCOMMS-20-27977A

First of all, we would like to appreciate the two reviewers very much for their valuable comments and suggestions on our manuscript with the code NCOMMS-20-27977A. The following is our point-by-point response to the referees.

Response to Reviewer #1

Comment: I appreciate the work the authors have done to improve the manuscript. The addition of calculations provide interesting insights regarding the magnetism. However, I still do not believe that these results meet the standard required for publication in Nature Communications. The observation of complicated charge ordering is certainly interesting, but better suited to a more specialized journal.

Response: We thank the referee to comment that the charge ordering observed in our manuscript is interesting. Actually, such a kind of fashion of charge ordering is quite unique, and it is never been found to occur in other materials. Moreover, the unique charge ordering of Pb ions leads to a peculiar alignment of Fe ions, giving rise to a spin reorientation transition at a critical temperature as high as 418 K. Compared with $R\text{FeO}_3$ (R = rare earth ions) perovskite family with anisotropic magnetic ions at the A site as the main origin of spin reorientation, new physical mechanism of spin reorientation is proposed in the current PbFeO_3 due to the absence of anisotropic rare earth ions. Therefore, the novel charge ordering is closely associated with new physics. As suggested by other two referees, these interesting results are deserved to be published in Nature Communications.

Response to Reviewer #2

Comment: The new theoretical insight into the mechanism of the spin-reorientation transition and its discussion based on the symmetry adapted distortion modes significantly improves the manuscript and I am happy to recommend the revised

version for publication in Nature Communication.

Response: We appreciate the referee very much for his/her recommendation to publish our revised manuscript in Nature Communications.